# Aortic carboxypeptidase-like protein potentiates β1 integrin signaling in mesenchymal progenitors

Cheyanne L Frosti, Diana Yeritsyan, Matthew D Layne

**Fibrosis is a pathological process characterized by persistent fibroblast activation and excessive ECM accumulation. Aortic carboxypeptidase-like protein (ACLP), a secreted ECM protein that binds fibrillar collagen, is up-regulated in fibrotic tissues and promotes fibroblast differentiation through canonical TGFβ receptor signaling. We hypothesized that when presented within the collagen matrix, ACLP would engage integrin-dependent mechanical signaling pathways that contribute to fibrogenic activation. Using 10T1/2 mouse mesenchymal progenitor cells, we identify a previously unrecognized mechanism through which collagen-bound ACLP induces fibrogenic activation via β1 integrin–mediated signaling. Collagen-bound ACLP induced rapid cell spreading, increased β1 integrin activation, and promoted focal adhesion maturation. These adhesion events triggered activation of the GTPases RhoA and Rac1, accompanied by enhanced F-actin assembly and nuclear accumulation of myocardin-related transcription factor A, a key regulator of fibrogenic gene expression. Transcriptomic profiling revealed enrichment of focal adhesion, ECM–receptor interaction, and actin cytoskeletal pathways downstream of collagen-bound ACLP, which was conserved in primary adipose-derived stromal cells. Together, these findings establish collagen-bound ACLP as a matrix-derived cue that links ECM composition to integrin-dependent fibrogenic activation.**

## Introduction

The ECM is a dynamic scaffold that regulates cell behavior during development, tissue repair, and disease progression through biochemical and mechanical cues (Doyle et al, 2022; Moretti et al, 2022; Di et al, 2023; Mierke, 2024). Disruption of ECM homeostasis is a common feature of pathological remodeling across diverse diseases, including fibrosis, cancer, cardiovascular disease, and heritable connective tissue disorders (Dzobo & Dandara, 2023; Yuan et al, 2023; Di Nubila et al, 2024; Salles Rosa Neto et al, 2024). Under these conditions, excessive matrix deposition and altered ECM organization increase tissue stiffness and perturb tissue architecture, often establishing feedback loops that further reinforce ECM remodeling and potentiate disease progression (Dooling et al, 2022; Lloyd & He, 2024).

Fibrosis is a form of pathological ECM remodeling across multiple organs including the lung, liver, kidney, heart, and adipose tissue (Zhao et al, 2022; Sun et al, 2023). It is characterized by excessive accumulation and crosslinking of ECM proteins, leading to tissue stiffening and progressive organ dysfunction (Yang & Plotnikov, 2021; Antar et al, 2023; Mayorca-Guiliani et al, 2025). Activated fibroblasts, or myofibroblasts, drive this process by producing fibrillar collagens such as collagen I (col1) and expressing contractile proteins including α-smooth muscle actin (αSMA) (Tomasek et al, 2002; Wen et al, 2022; Sharip & Kunz, 2025). Fibroblast activation is regulated by both biochemical signals, such as transforming growth factor beta (TGFβ), and mechanical cues derived from the ECM (Hinz, 2009; Piersma et al, 2015; Zhao et al, 2022). These mechanical signals engage integrins and remodel the actin cytoskeleton, activating mechanosensitive transcriptional regulators including myocardin-related transcription factor A (MRTFA) that cooperate with TGFβ signaling to reinforce fibrotic gene expression (Liu et al, 2010; Piersma et al, 2015; Santos & Lagares, 2018; Di et al, 2023; Sharip & Kunz, 2025).

In white adipose tissue (WAT), fibrotic remodeling is increasingly recognized as a pathogenic feature of obesity and metabolic disease, where excessive collagen deposition limits adipose tissue expandability and contributes to insulin resistance and chronic inflammation (Sun et al, 2013; Marcelin et al, 2022). WAT contains a heterogeneous stromal vascular fraction (SVF) composed of mesenchymal progenitor cells, endothelial cells, immune cells, and perivascular populations that collectively regulate adipose tissue remodeling (Tang et al, 2008; Merrick et al, 2019). Among these populations, adipose-derived stromal progenitors retain the capacity to differentiate toward adipogenic lineages that support tissue expansion or toward fibrogenic fibroblast-like states that promote ECM deposition and fibrosis (Hepler et al, 2018). Signals within the adipose ECM are therefore key regulators of stromal progenitor fate decisions, determining whether adipose tissue undergoes healthy expansion through adipogenesis or pathological remodeling characterized by fibrosis.

Department of Biochemistry and Cell Biology, Boston University Chobanian and Avedisian School of Medicine, Boston, MA, USA

Correspondence: mlayne@bu.edu

Aortic carboxypeptidase-like protein (ACLP), encoded by the *AEBP1* gene, is a secreted ECM protein that was originally identified as a vascular smooth muscle differentiation marker (Layne et al, 1998, 2002). ACLP is induced during vascular injury, wound repair, and neointima formation (Layne et al, 2001; Ith et al, 2005; Blackburn et al, 2018; Syx et al, 2019); and is up-regulated in collagen-rich, actively remodeling tissues including embryonic mesenchyme (Ith et al, 2005), fibrotic lung, liver, heart, and adipose tissues (Schissel et al, 2009; Tumelty et al, 2014; Jager et al, 2018; Teratani et al, 2018; Wang et al, 2021; Kattih et al, 2023; Zhang et al, 2023), as well as in the tumor stroma (Sekiguchi et al, 2023; Wang et al, 2025). Several studies have shown that ACLP is a potent profibrotic soluble factor that promotes fibroblast activation through TGFβ receptor I–dependent and TGFβ receptor I–independent mechanisms (Gagnon et al, 2005; Tumelty et al, 2014; Jager et al, 2018; Kattih et al, 2023) and can act as a noncanonical Wnt ligand in hepatic stellate cells (Teratani et al, 2018). In WAT, ACLP is strongly induced in fibrotic WAT depots and promotes stromal progenitor cell activation while suppressing adipogenic differentiation, linking ACLP expression to pathological adipose tissue remodeling (Jager et al, 2018). Together, these findings identify ACLP as an ECM-associated factor capable of regulating stromal progenitor cell behavior in fibrotic tissues.

ACLP binds fibrillar collagen through its discoidin domain and alters collagen fiber mechanics (Blackburn et al, 2018; Vishwanath et al, 2020). Loss-of-function *AEBP1* variants in humans impair collagen organization and ECM integrity causing the connective tissue disorder Ehlers–Danlos syndrome (Blackburn et al, 2018; Angwin et al, 2023). In collagen-rich matrices, ACLP increases fibroblast proliferation and contractility (Schissel et al, 2009), and in vascular adventitial cultures, matrix-associated ACLP promotes MRTFA-dependent fibroblast differentiation (Wang et al, 2021), indicating that collagen-associated ACLP can influence fibroblast phenotype. Despite these observations, the mechanisms by which ACLP initiates intracellular signaling when associated with the collagen matrix remain poorly understood. Prior studies demonstrate that soluble ACLP activates canonical profibrotic pathways, yet it is not known whether collagen-bound ACLP engages distinct mechanical signaling pathways rather than acting solely through mechanisms shared with its soluble form.

Here, we investigated whether collagen-bound ACLP activates early profibrotic mechanical responses and examined how these pathways relate to canonical TGFβRI signaling. Using collagen-coated hydrogels of defined stiffness, we defined the earliest signaling events triggered by collagen-bound ACLP. Our findings demonstrate that ACLP functions as an ECM-derived mechanical signal that activates β1 integrins, enhances cytoskeletal organization, and drives MRTFA nuclear localization, establishing a previously unrecognized pathway by which collagen-bound ACLP promotes fibrogenic activation.

# Results

## Early cell spreading is enhanced by collagen-bound ACLP through TGFβR1-independent mechanisms

To study early cellular responses to ACLP, we generated recombinant ACLP from mammalian cells (Tumelty et al, 2014),

verified its purity by SDS–PAGE (Fig S1A), and confirmed identity by immunoblotting using a myc-tag antibody (Fig S1B). Using a previously established cell-free collagen polymerization assay, ACLP increased collagen fibrillogenesis (Fig S1C), consistent with prior reports and validating its functional activity (Blackburn et al, 2018).

Cells that adopt fibroblast-like phenotypes are highly sensitive to substrate mechanics, and on tissue culture plastic, they rapidly adopt a persistently activated, myofibroblast-like phenotype through mechanosensitive pathways (Discher et al, 2005; Sen et al, 2009). To control substrate stiffness while isolating ECM-specific effects, we cultured cells on 12 kPa polyacrylamide hydrogels coated with type I collagen polymerized in the presence or absence of ACLP, a copolymerization approach that presents ACLP in a matrix-associated state and reflects its physiological localization within remodeling ECM, where ACLP colocalizes with fibrillar collagen (Ith et al, 2005; Tumelty et al, 2014; Vishwanath et al, 2020). Collagen-associated retention of ACLP after copolymerization was assessed by an ELISA-based assay (Fig S1D). Although this approach does not directly demonstrate physical incorporation, ACLP was retained only when present during collagen polymerization and not when added in soluble form after gel formation. Furthermore, ACLP was not retained on gelatin (denatured collagen) substrates, indicating collagen-specific matrix association. These findings are consistent with prior studies from our laboratory demonstrating ACLP incorporation into engineered collagen fibers (Vishwanath et al, 2020) and detection of ACLP within the ECM fraction of cell-excreted collagen matrices (Schissel et al, 2009). The following experiments were performed using 10T1/2 cells, a well-established mouse mesenchymal progenitor cell line that exhibits fibroblast-like morphology, ECM production, and robust responsiveness to profibrotic signaling, making it a useful in vitro model for studying fibrogenic activation pathways (Pinney & Emerson, 1989; Putra et al, 2023).

One of the earliest morphological changes associated with fibrogenic activation is cell spreading, a sensitive readout of cytoskeletal engagement and mechanotransduction (Yeung et al, 2005; Chan & Odde, 2008). To determine whether collagen-bound ACLP initiates this response, we quantified cell spreading in 10T1/2 mesenchymal progenitors. Cells were serum-starved in suspension and seeded onto hydrogels coated with collagen alone (col1) or collagen copolymerized with ACLP (col1-ACLP) (Fig 1A). Across a 30- to 90-min time course, col1-ACLP significantly increased cell area relative to col1 controls (Fig 1B), and this difference persisted at 48 h (Fig S2A). In contrast, soluble ACLP failed to increase cell spreading at 90 min, indicating that matrix association is required for this response (Fig 1C). The number of adherent cells also did not change, indicating that ACLP enhances cell spreading rather than initial adhesion (Fig S2B).

Previous studies have shown that soluble ACLP activates Smad2/3 phosphorylation through TGFβR1 (Tumelty et al, 2014), and TGFβR1 inhibition attenuates ACLP-induced αSMA expression during adipogenic differentiation (Jager et al, 2018), suggesting canonical TGFβR1 signaling can mediate ACLP's profibrotic activity. We therefore asked whether the early spreading response to col1-ACLP was dependent on TGFβR1 kinase activity. Cells treated with the TGFβR1/ALK5 inhibitor, SB431542, during seeding (col1-ACLP+SB) exhibited increased cell spreading to the same extent as

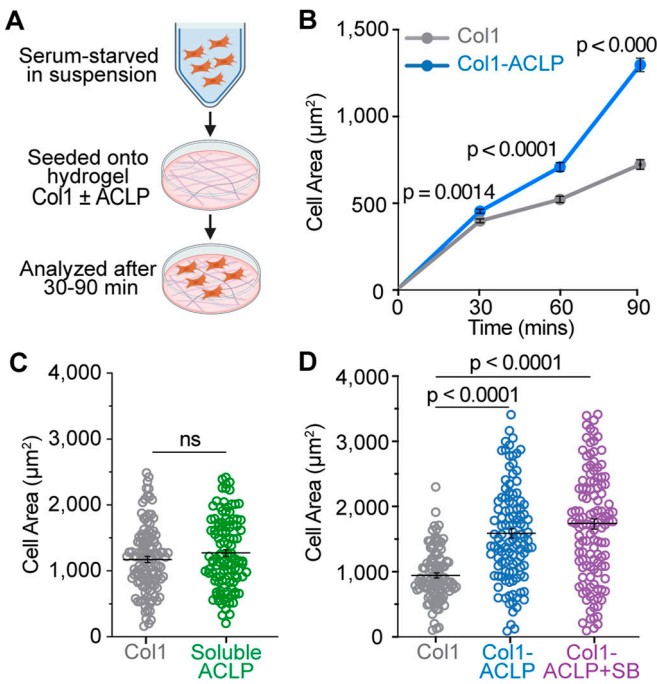

Figure 1. Collagen-bound aortic carboxypeptidase-like protein (ACLP) induces early cell spreading through TGFβR1-independent mechanisms. **(A)** Schematic of the cell stimulation by the ECM assay. 10T1/2 mesenchymal progenitors were serum-starved in suspension for 1 h, then seeded onto 12 kPa polyacrylamide hydrogels coated with type I collagen polymerized ± 30 nM recombinant ACLP. **(B)** Cells were fixed at 30, 60, and 90 min and stained with phalloidin (Alexa Fluor 555) to visualize F-actin. Phalloidin staining was segmented in CellProfiler using minimum cross-entropy thresholding (diameter 100–100,000 px), and cell area was measured from the resulting single-cell objects. **(C)** Cell spreading at 90 min on col1 hydrogels treated with 30 nM soluble ACLP. **(D)** Cell spreading at 90 min ± 5 $\mu$M SB431542 (TGFβR/ALK5 inhibitor). Data represent the mean ± SD from three independent experiments (n = 3). At least 30 cells per condition per replicate were analyzed (≥100 cells total). Statistical comparisons used Welch's t tests; P < 0.05.

cells on col1-ACLP matrices alone (Fig 1D). These results demonstrate that the rapid morphological activation driven by collagen-bound ACLP occurs independent of TGFβR1 signaling, identifying ACLP as a matrix-embedded cue that initiates early fibrogenic activation.

## Collagen-bound ACLP enhances β1 integrin activation and focal adhesion maturation

Because integrins transmit ECM-based cues into intracellular signaling, we next asked whether collagen-bound ACLP regulates β1 integrin activation and focal adhesion complexes. β1 integrins are established collagen-binding receptors (Hynes, 2002; Sun et al, 2016), and their transition from an inactive conformation to an active conformation is a critical step in transmitting ECM cues into intracellular adhesion and signaling complexes (Liu et al, 2009; Campbell & Humphries, 2011). Because the signaling events examined here, including integrin conformational activation and focal adhesion clustering, depend on protein localization and conformational state rather than total protein abundance, we employed quantitative, single-cell immunofluorescence approaches

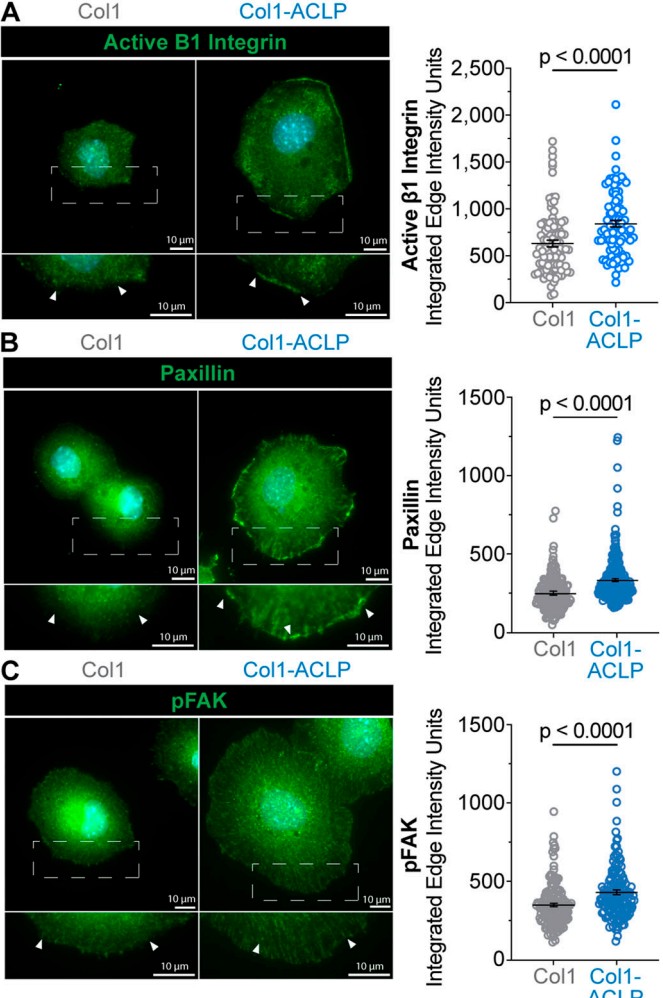

Figure 2. Collagen-bound aortic carboxypeptidase-like protein (ACLP) enhances β1 integrin activation and focal adhesion maturation.
10T1/2 mesenchymal progenitors were seeded on col1 or col1-ACLP hydrogels as described in Fig 1A and fixed after 90 min. **(A, B, C)** Cells were stained for (left) (A) active β1 integrin (HUTS-4), (B) paxillin, or (C) pFAK (Tyr397). Representative images are shown; scale bars, 10 $\mu$m. Quantification (right) of integrated fluorescence edge intensity in CellProfiler. Nuclei were segmented (minimum cross-entropy), used to define single-cell boundaries, and signal intensities were measured from the corresponding fluorescence channels. Per-cell values represent integrated edge intensity measurements. Data represent the mean ± SD from three independent experiments (n = 3). At least 30 cells per condition per replicate were analyzed (≥100 cells in total). Statistical comparisons were performed using Welch's t tests; P < 0.05.

to assess pathway activation. Using an antibody that recognizes the active conformation of β1 integrin (9EG7 clone), cells cultured on col1-ACLP exhibited significantly greater activation of β1 integrins compared with col1 controls (Fig 2A).

To assess whether increased β1 integrin activation translated into enhanced adhesion signaling, we next examined two downstream focal adhesion components, paxillin and focal adhesion kinase (FAK). Paxillin, which coordinates the assembly and maturation of nascent adhesions (Zaidel-Bar et al, 2007), and FAK autophosphorylation at Tyr397, an early integrin-driven signaling event, together mark focal adhesion activation and maturation

(Zhao et al, 2016). Consistent with increased integrin engagement, paxillin-positive focal adhesions were more abundant on col1-ACLP matrices, showing greater periphery enrichment where new force-bearing focal adhesions typically assemble (Fig 2B). Levels of pFAK were similarly elevated in cells on col1-ACLP relative to col1, supporting enhanced integrin signaling (Fig 2C). Together, these data indicate that collagen-bound ACLP increases $\beta1$ integrin activation and promotes focal adhesion assembly and maturation, key steps in initiating integrin-mediated mechanotransduction in mesenchymal progenitor cells undergoing fibrogenic activation.

## Collagen-bound ACLP activates RhoA and Rac1 signaling

Integrin engagement with the ECM activates Rho family GTPases, with RhoA driving stress fiber formation and contractility, Rac1 promoting lamellipodial spreading, and Cdc42 mediating filopodium assembly (Burridge & Wennerberg, 2004; Heasman & Ridley, 2008). These GTPases serve as critical effectors that link ECM cues to cytoskeleton remodeling and cell behavior (Liu et al, 2008; Xu et al, 2009; Hall, 2012). Because collagen-bound ACLP enhanced $\beta1$ integrin activation and focal adhesion maturation, we next asked whether ACLP augments downstream activation of Rho family GTPases. Cells were serum-starved in suspension, with a time zero (T0) aliquot collected as baseline, and then seeded on col1, col1-ACLP, or col1-ACLP+SB hydrogels. As expected, cells on col1 displayed increased RhoA and Rac1 activity compared with T0, reflecting integrin–collagen engagement. However, cells on col1-ACLP exhibited significantly higher Rac1 and RhoA activation than col1 controls (Fig 3A and B), whereas Cdc42 activity was reduced across conditions (Fig 3C). These results persisted with inhibition of ALK5 (col1-ACLP+SB) and were consistent with actin cytoskeleton features observed on col1-ACLP, where cells spread but displayed no filopodia-like protrusions. Together, these findings demonstrate that collagen-bound ACLP enhances activation of RhoA and Rac1, supporting a role for ACLP in amplifying integrin-mediated signaling.

## Collagen-bound ACLP promotes F-actin assembly and enhances MRTFA nuclear translocation

Integrin engagement reorganizes the actin cytoskeletal network through Rho family GTPases, driving stress fiber formation and increasing cell spreading (Lawson & Burridge, 2014). Stress fiber assembly is a defining feature of fibrogenic activation and is closely linked to mesenchymal progenitor differentiation into activated fibroblasts (Hinz, 2010). Because our earlier results showed ACLP enhanced spreading and RhoA/Rac1 activity, we next tested whether collagen-bound ACLP would promote actin polymerization. Cells cultured on col1-ACLP displayed significantly greater phalloidin-stained filamentous actin (F-actin) compared with col1 controls, with more prominent stress fibers (Fig 4A). This increase was maintained in the presence of the ALK5 inhibition (col1-ACLP+SB), confirming that ACLP-driven actin assembly occurs independent of TGF$\beta$R1 signaling. Consistent with these findings, G/F-actin fractionation assays revealed a higher ratio of poly-merized F-actin to globular actin (G-actin) on col1-ACLP (Fig 4B). Although collagen engagement alone induced basal stress fiber formation, incorporation of ACLP into the matrix further enhanced

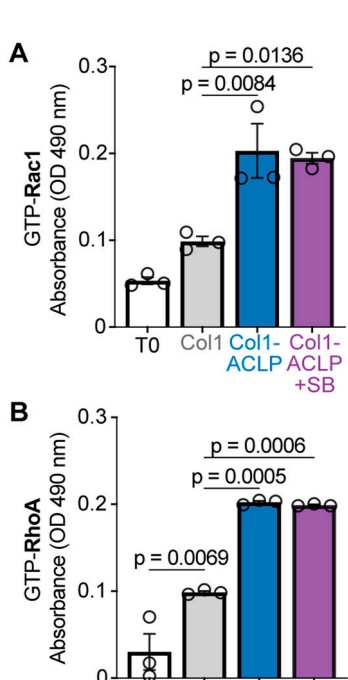

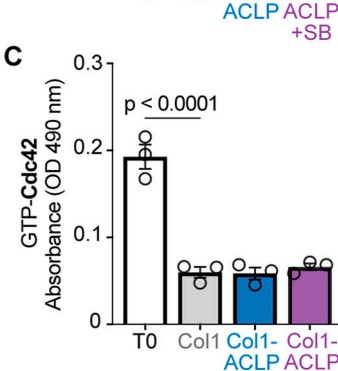

**Figure 3. Collagen-bound aortic carboxypeptidase-like protein (ACLP) enhanced RhoA and Rac1 GTPase activity.**
10T1/2 mesenchymal progenitors were seeded on col1, col1-ACLP, or col1-ACLP+SB hydrogels for 30 min as described in Fig 1A. **(A, B, C)** Activation of (A) Rac1, (B) RhoA, and (C) Cdc42 was measured using G-LISA activation assays (Cytoskeleton, Inc.). Absorbance was measured at 490 nm. Data represent the mean ± SD from three independent experiments ($3 \times 10^5$ cells per n; n = 3). Statistical comparisons were performed using Welch's $t$ tests or one-way ANOVA, as appropriate; $P < 0.05$.

this response, indicating that ACLP strengthens the cytoskeletal architecture associated with early fibrogenic activation.

Actin polymerization regulates gene expression through several mechanisms including the release of MRTFA from G-actin, enabling its nuclear translocation and coactivation of SRF-dependent transcription (Small, 2012; Haak et al, 2014). This actin/MRTFA axis is a well-established route by which matrix stiffness and integrin signaling influence profibrotic transcriptional programs (Small, 2012; Shiwen et al, 2015; Fearing et al, 2019; Yang & Plotnikov, 2021). We therefore examined MRTFA localization in cells cultured on col1-ACLP matrices. Immunofluorescence imaging revealed that cells on col1-ACLP exhibited significantly higher nuclear MRTFA

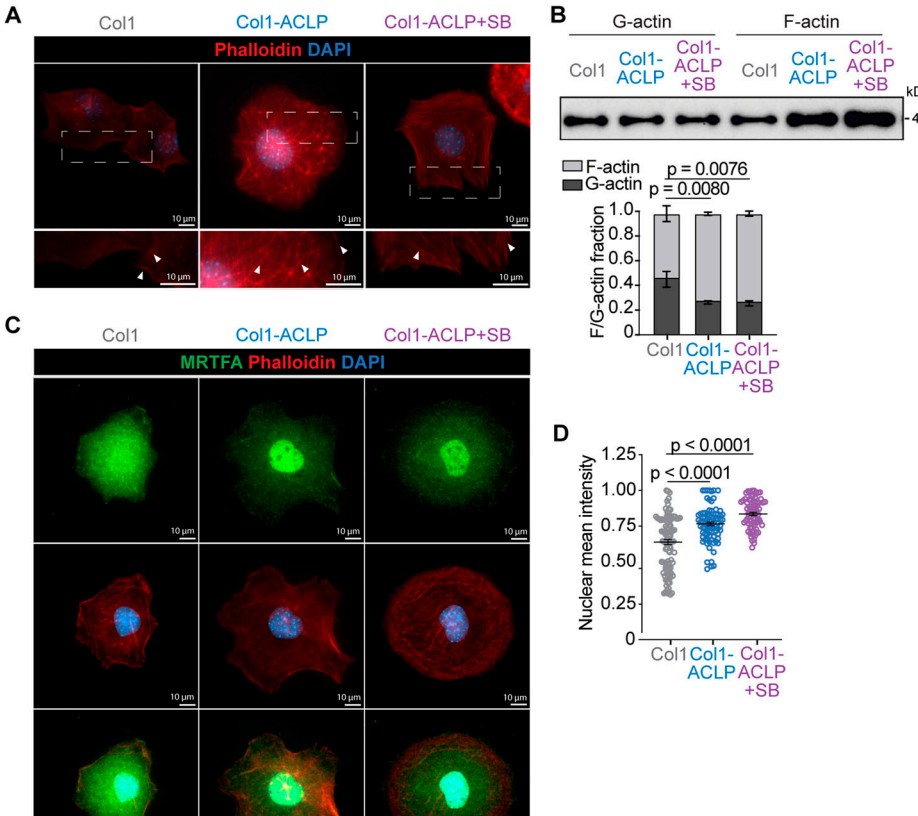

**Figure 4. Collagen-bound aortic carboxypeptidase-like protein (ACLP) increases F-actin assembly and MRTFA nuclear accumulation.**
10T1/2 mesenchymal progenitors were seeded on col1, col1-ACLP, or col1-ACLP+SB hydrogels and fixed after 90 min. **(A)** Immunofluorescence staining of F-actin with phalloidin (Alexa Fluor 555). Representative images are shown; scale bars, 10 $\mu$m. **(B)** (left) Immunoblot of G-actin and F-actin fractions separated using a G/F-actin assay kit (Cytoskeleton, BK037). **(B)** (Right) Quantification of the G-actin/F-actin ratio from three independent experiments ($3 \times 10^5$ cells per n; n = 3). **(C)** Immunofluorescence staining for MRTFA (Alexa Fluor 488) with phalloidin and DAPI nuclear counterstain. Representative images are shown; scale bars, 10 $\mu$m. **(D)** Quantification of MRTFA localization in CellProfiler. Nuclei and phalloidin were segmented (minimum cross-entropy), used to define nuclear and cytoplasmic boundaries, and signal intensities were measured from the corresponding fluorescence channels. Data represent the mean ± SD from three independent experiments. At least 30 cells per condition per replicate were analyzed (≥100 cells in total). Statistical comparisons were performed using Welch's $t$ test or one-way ANOVA as appropriate; $P < 0.05$.

intensity compared with col1 controls (Fig 4C and D), and this increase persisted with ALK5 inhibition (col1-ACLP+SB). These results show that cells cultured on collagen-bound ACLP display increased MRTFA nuclear localization, consistent with enhanced actin remodeling under fibrogenic activation.

## Collagen-bound ACLP activates a mechanosensitive, activated fibroblast transcriptional response

Having defined a signaling pathway in which collagen-bound ACLP enhances $\beta$1 integrin activation, RhoA/Rac1 signaling, actin assembly, and MRTFA nuclear localization, we next asked whether ACLP elicits a corresponding transcriptional program and whether this response persists with ALK5 inhibition. To capture early transcriptional events, 10T1/2 mesenchymal progenitor cells were cultured for 18 h on col1, col1-ACLP, or col1-ACLP+SB hydrogels. Bulk RNA sequencing followed by principal component analysis showed separation of the three conditions, indicating distinct transcriptional states (Fig 5A). Differential expression analysis identified a substantial ACLP-responsive gene set (Table S1), reflecting strong early transcriptional events downstream of ACLP. To validate the findings, we quantified the expression of representative differentially expressed genes (DEGs) between the three conditions and observed consistency with the RNA-seq (Fig S3).

A volcano plot comparing col1-ACLP with col1 revealed induction of profibrotic and matrix remodeling genes such as *Col8a1*, *Lox*, *Thbs1*, *Acta2*, *Serpine1*, and *Col1a1* (Fig 5B). Collagen-bound ACLP also suppressed transcripts associated with stromal progenitor-like or quiescent states, including *Pdgfra*, *Gfra1*, *Ebf1*, *Akap12*, and *Pdk4*, as well as ECM-organizing or anti-fibrotic regulators such as *Dcn*, *Col18a1*, *Angpt1*, and *Sned1*, consistent with a shift toward an activated fibroblast-like phenotype.

Pathway enrichment analysis using the Kyoto Encyclopedia of Genes and Genomes (KEGG) database revealed significant up-regulation of mechanosensitive and ECM-associated pathways, including focal adhesion, regulation of actin cytoskeleton, and ECM–receptor interaction (Fig 5C; Table S2). Enrichment across these pathways was driven by up-regulation of genes involved in cell adhesion and integrin activation (*Itgb1*, *Itga11*, *Itga6*, *Itgav*, *Cav1*, *Cav2*, *Crk*), cytoskeletal organization and contractility (*Actn1*, *Actn4*, *Myl12a*, *Myh9*, *Rhoa*, *Rac1*, *Cdc42*, *Iqgap1*, *Dock1*, *Pak3*, *Pak4*, *Limk2*), and matrix assembly and crosslinking (*Col1a1*, *Col4a1*, *Col4a2*, *Fn1*, *Tnc*, *Lox*) (Fig 5D). Collagen-bound ACLP also increased enrichment of pathways related to proliferation and matrix remodeling, including PI3K-Akt signaling and cell-cycle regulation, reflecting early shifts in mesenchymal progenitor differentiation into activated fibroblasts. The pathways up-regulated at the transcript level align with the upstream signaling events, suggesting potential reinforcing or perpetuating mechanisms common in fibrosis, and together, these

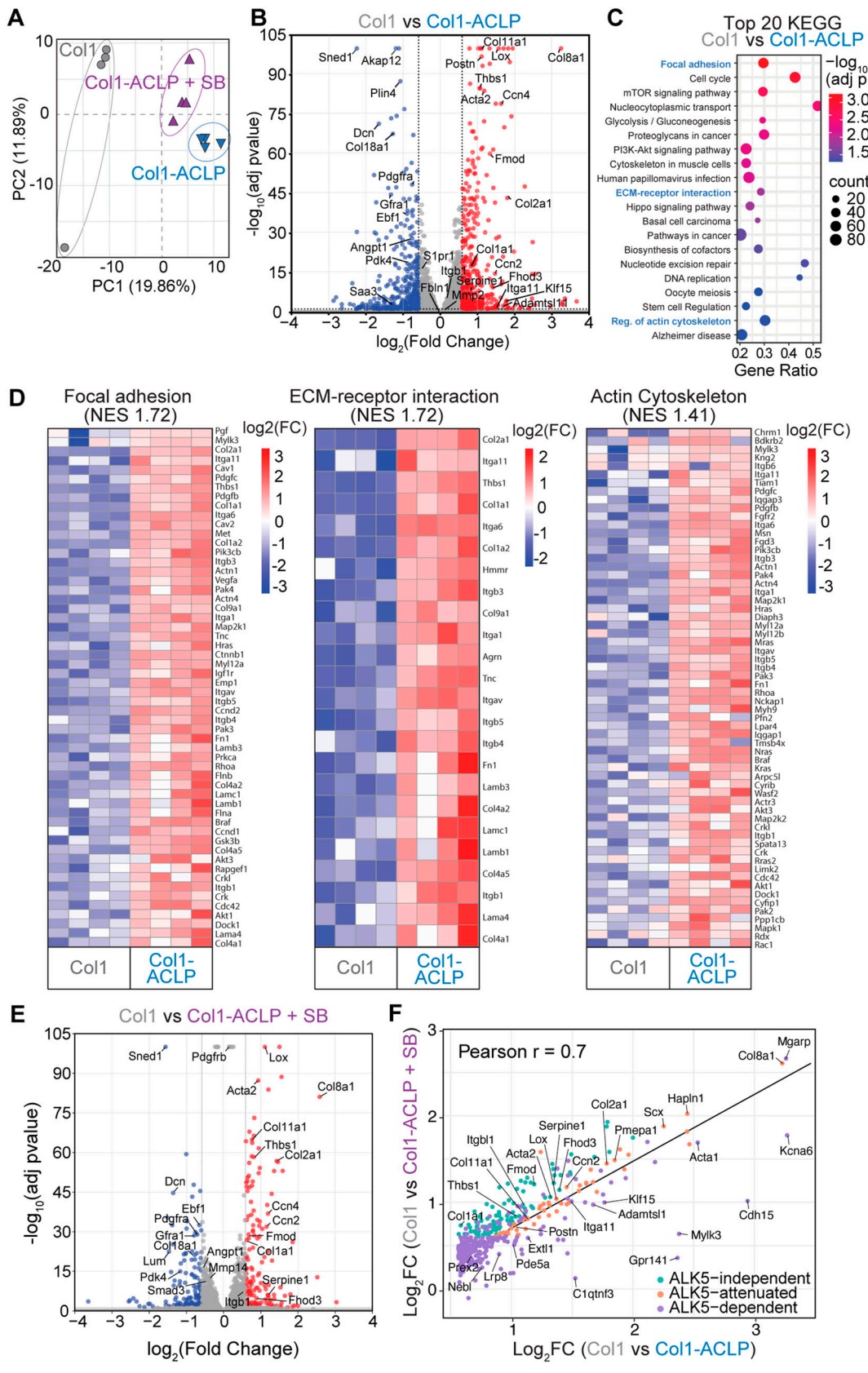

responses identify collagen-bound ACLP as a potent ECM-derived cue that activates an early mechanosensitive, profibrotic transcriptional program.

### Collagen-bound ACLP activates a predominately TGFβR1-independent transcriptional program linked to integrin-mediated signaling

We next examined the transcriptional response induced by ACLP when ALK5 signaling was inhibited at the time of plating (col1-ACLP+SB), to determine which ACLP-driven gene expression changes persist in the absence of canonical TGFβR1 activity. The volcano plot comparing col1-ACLP+SB with col1 revealed continued induction of many mechanosensitive and profibrotic targets, including *Lox, Acta2, Col8a1, Ccn2, Ccn4,* and *Serpine1,* whereas transcripts such as *Sned1, Smad3, Dan, Angpt1,* and *Mmp14* were reduced (Fig 5E). Several canonical TGFβ-responsive genes, including *Serpine1, Hey1,* and *Skil,* remained partially induced, consistent with ACLP-activating integrin- and cytoskeleton-driven pathways that converge on transcriptional outputs often ascribed to TGFβ signaling (Piersma et al, 2015).

To determine how ALK5 inhibition modified ACLP-dependent gene expression, we compared $\log_2$ fold changes between col1 versus col1-ACLP and col1 versus col1-ACLP+SB and grouped up-regulated genes by the extent to which ALK5 inhibition altered their expression (Fig 5F; Table S3). A large subset of ACLP-induced genes showed minimal change with ALK5 inhibition, indicating ALK5 independence; a second subset remained up-regulated but at a reduced magnitude, reflecting ALK5 attenuation; and a smaller group fell below our $\log_2$FC cutoff when ALK5 was inhibited, consistent with ALK5 dependency. Across these categories, ALK5-independent and ALK5-attenuated genes were enriched for ECM assembly, integrin–cytoskeletal remodeling, and inflammatory mediators, closely matching the mechanical signaling pathways identified earlier. In contrast, ALK5-dependent genes were dominated by cell-cycle and mitotic regulators, along with a smaller subset of matrix-modifying factors. Together, these patterns indicate that collagen-bound ACLP engages both ALK5-dependent and ALK5-independent gene programs, with the dominant transcriptional response reflecting integrin–cytoskeletal mechanical signaling.

### Collagen-bound ACLP promotes mechanosensitive transcriptional programs in primary stromal progenitors

Given prior evidence implicating ACLP in adipose tissue fibrosis and stromal remodeling in vivo (Jager et al, 2018) and the capacity of 10T1/2 mesenchymal progenitors to commit to a fibrogenic or adipogenic lineage, we next assessed whether collagen-bound ACLP elicits similar transcriptional responses in primary adipose-derived stromal progenitors isolated from the stromal vascular fraction (SVF). The SVF contains a heterogeneous population of stromal cells, including mesenchymal progenitors, capable of adopting both fibrogenic and adipogenic lineages. SVF cells isolated from mouse gonadal white adipose tissue (gWAT) were plated on 12 kPa polyacrylamide hydrogels coated with type I collagen alone (col1) or collagen polymerized in the presence of ACLP (col1-ACLP) and cultured for 48 h (Fig 6A).

Bulk RNA sequencing followed by principal component analysis revealed clear separation between col1-ACLP and col1 conditions, indicating that exposure to collagen-bound ACLP broadly alters the transcriptional state of primary cells (Fig 6B). Notably, several core profibrotic genes identified in 10T1/2 cells, including *Postn, Serpine1,* and *Ccn2,* were similarly up-regulated in primary cells, indicating conservation of ACLP-responsive transcriptional programs across model systems (Fig 6C). The transcriptional response to collagen-bound ACLP was next examined to determine whether ACLP preferentially enriched specific gene expression programs. Pathway enrichment analysis demonstrated selective enrichment of ECM- and mechanosensitive transcriptional pathways downstream of col1-ACLP, including focal adhesion, ECM–receptor interaction, and regulation of actin cytoskeleton (Fig 6D). These enrichments were driven by coordinated up-regulation of genes encoding integrins (*Itga5, Itga11, Itgb1, Itgb5*), cytoskeletal regulators (*Actn1, Myl9, Iqgap1, Limk2*), and ECM structural components (*Col1a1, Col1a2, Fn1, Tnc, Lama1*), consistent with transcriptional programs associated with matrix engagement and force-responsive adaptation. Together, these results indicate that collagen-bound ACLP elicits conserved transcriptional responses in primary progenitors and preferentially biases stromal gene expression toward programs associated with ECM remodeling and mechanosensitive adaptation.

## Discussion

This study identified collagen-bound ACLP as a profibrotic ECM regulator that signals through an integrin-mediated pathway. When incorporated into collagen, ACLP promoted β1 integrin activation, RhoA/Rac1 signaling, actin polymerization, and MRTFA nuclear translocation, hallmarks of mechanosensitive fibrogenic activation. These responses persisted with

---

**Figure 5. Aortic carboxypeptidase-like protein (ACLP) activates a transcriptional program revealing enriched for focal adhesions, ECM– receptor interactions, and actin cytoskeleton pathways.**
**(A)** Principal component analysis of bulk RNA-seq data from 10T1/2 mesenchymal progenitors cultured for 18 h on col1, col1-ACLP, or col1-ACLP+SB hydrogels (n = 4 biological replicates per group). **(B)** Volcano plot showing differentially expressed genes (DEGs) between col1 and col1-ACLP. Significantly up-regulated genes ($\log_2$FC > 0.58, adj $P$ < 0.05) are shown in red and significantly down-regulated genes ($\log_2$FC < −0.58, adj $P$ < 0.05) in blue; nonsignificant genes appear in gray. The dashed horizontal line denotes $-\log_{10}$ (adj $P$ = 0.05). **(C)** KEGG GSEA of genes up-regulated in col1-ACLP relative to col1. Pathways associated with focal adhesion, regulation of actin cytoskeleton, and ECM–receptor interaction were significantly enriched (FDR < 0.05). **(C, D)** Heatmaps of core gene enrichments from selected KEGG pathways shown in (C), displayed as $\log_2$FC relative to col1. **(E)** Volcano plot comparing col1 and col1-ACLP+SB. **(B)** Significantly up-regulated and down-regulated genes are colored as in (B). **(F)** Correlation analysis of per-gene $\log_2$FC comparing col1 versus col1-ACLP (x-axis) and col1 versus col1-ACLP+SB (y-axis). Pearson's correlation coefficient r is shown. Genes are color-coded by ALK5 dependence: ALK5-independent (up-regulated in both conditions; |Δ $\log_2$FC| < 0.3), ALK5-attenuated (up-regulated in both with reduced magnitude in SB condition; |Δ $\log_2$FC| ≥ 0.3, adj $P$ < 0.05), and ALK5-dependent (significant with col1-ACLP but not with SB).

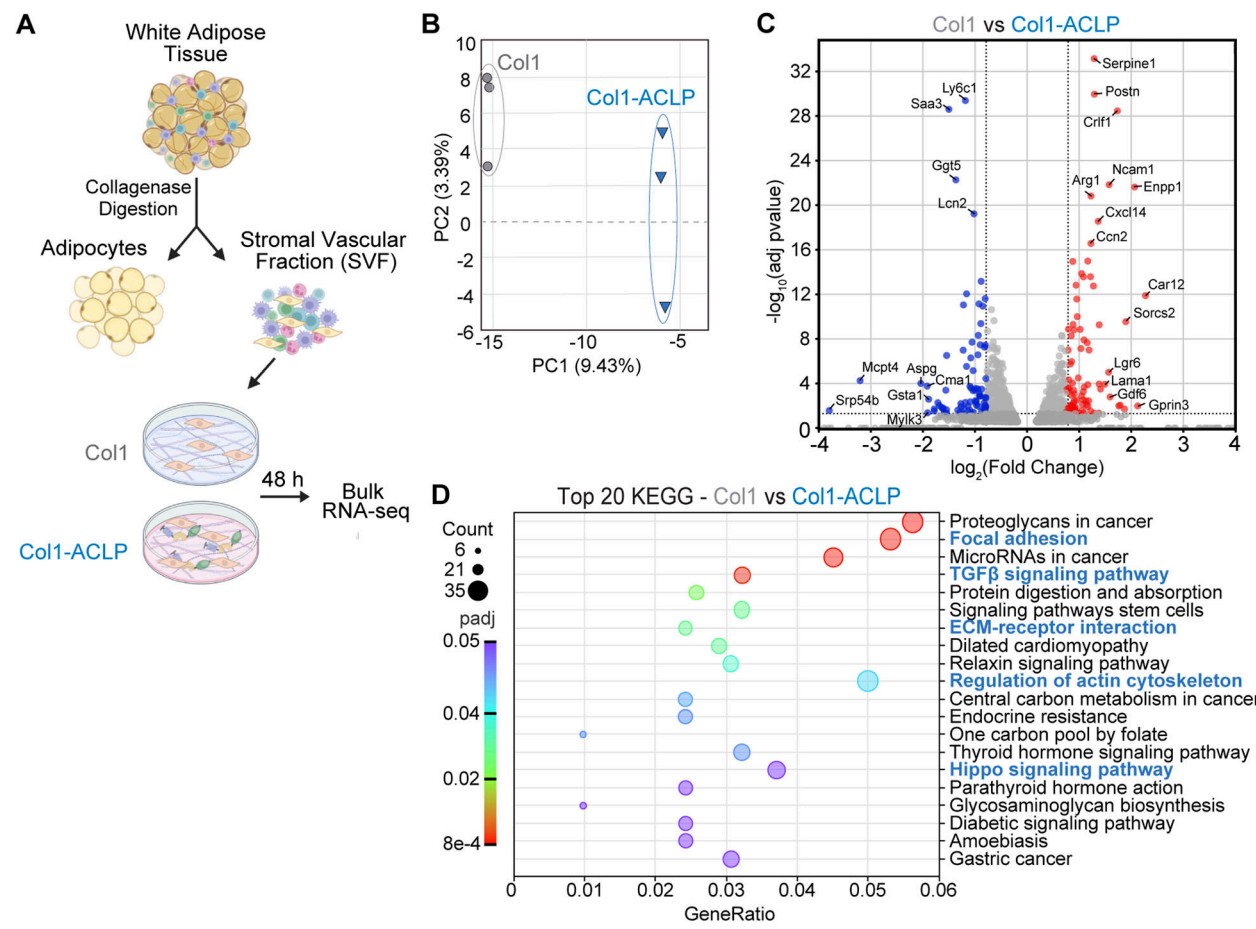

**Figure 6. Collagen-bound aortic carboxypeptidase-like protein (ACLP) elicits conserved transcriptional responses in primary stromal progenitors.**
**(A)** Gonadal white adipose tissue (gWAT) was digested to isolate the stromal vascular fraction (SVF). Cells were plated for 48 h on 12 kPa hydrogels coated with type I collagen polymerized ± 30 nM ACLP. **(B)** Principal component analysis of bulk RNA-seq data (n = 3 biological replicates per condition). **(C)** Volcano plot showing differentially expressed genes (DEGs) between col1 and col1-ACLP. Significantly up-regulated genes ($\log_2$FC > 0.58, adj $P$ < 0.05) are shown in red and significantly down-regulated genes ($\log_2$FC < −0.58, adj $P$ < 0.05) in blue; nonsignificant genes appear in gray. Dashed horizontal line denotes −$\log_{10}$ (adj $P$ = 0.05). **(D)** KEGG GSEA of genes up-regulated in col1-ACLP relative to col1. Pathways associated with focal adhesion, regulation of actin cytoskeleton, and ECM–receptor interaction were significantly enriched (FDR < 0.05).

ALK5 inhibition, establishing ACLP as a matrix-derived cue capable of sustaining fibrogenic activation independent of canonical TGFβR1 signaling.

## Convergence of integrin-mediated and soluble ACLP signaling pathways

Fibrosis is a maladaptive tissue repair response driven by persistent fibroblast activation and excessive ECM deposition. Although TGFβ is a central driver of this process, biochemical cues arising from ECM composition have an equally central role (Hinz, 2009; Piersma et al, 2015). Prior studies from our group and others have established soluble ACLP as a profibrotic factor, functioning through the TGFβR1 complex to increase Smad2/3 phosphorylation (Tumelty et al, 2014; Jager et al, 2018; Kattih et al, 2023). In addition to canonical Smad2/3 activation, soluble ACLP also engages Smad1/5/9 signaling, consistent with BMP-responsive pathways (Tumelty et al, 2014), and noncanonical Wnt signaling (Teratani

et al, 2018), underscoring its ability to participate in multiple profibrotic signaling networks. Our findings extend this model by showing that ACLP, when bound to collagen, activates β1 integrins and the downstream RhoA/MRTFA axis, identifying a novel profibrotic, integrin-mediated signaling pathway. Although this study focused on collagen-bound ACLP, prior work on soluble ACLP suggests that these biochemical and mechanical signaling modes likely converge in vivo. Integrin engagement can facilitate latent TGFβ activation (Wipff et al, 2007), TGFβ can activate RhoA to enhance cytoskeletal tension and influence Smad nuclear dynamics (Moustakas & Heldin, 2005), and MRTFA and Smad proteins coregulate overlapping transcriptional programs (Small, 2012). Such points illustrate how biochemical and mechanical pathways can act in parallel or in compensatory ways within fibrotic microenvironments, positioning ACLP as a dual-mode ECM cue capable of integrating biochemical and mechanical signals that reinforce fibrogenic activation.

### Integration of ACLP structural roles with mechanical signaling

ACLP's ability to drive integrin-mediated signaling aligns with its known structural and mechanical roles within collagen matrices. The C-terminal discoidin domain binds fibrillar collagen (Blackburn et al, 2018) and modifies collagen fiber organization and mechanical properties (Vishwanath et al, 2020). In collagen-rich environments, ACLP increases fibroblast proliferation, contractility, and matrix remodeling capacities, effects observed in human lung fibroblasts cultured within collagen gels (Schissel et al, 2009) and in vascular adventitial fibroblasts, where ACLP promotes MRTFA-dependent differentiation (Wang et al, 2021). Our findings build on this work by demonstrating that collagen-associated ACLP not only influences fibroblast phenotype but also directly enhances β1 integrin activation and focal adhesion maturation, providing a mechanistic link between ACLP's matrix incorporation and integrin-mediated signaling. By defining this upstream integrin–RhoA/Rac1–cytoskeleton pathway, our results connect ACLP's structural presence in collagen to the mechanical signals that position MRTFA for nuclear accumulation and early transcriptional activation. Together, these findings position ACLP as a structural ECM protein capable of modulating both matrix mechanics and integrin-mediated mechanotransduction.

### Presentation state–dependent signaling of ACLP in fibrotic microenvironments

Collagen-bound ACLP promoted rapid cell spreading, whereas soluble ACLP did not increase cell spreading on collagen matrices, indicating that matrix association is required for ACLP to support force-dependent signaling. Similar presentation-dependent regulation has been described for other matricellular proteins. Tenascin-C modulates integrin engagement and mechanotransduction when embedded within fibronectin-rich matrices, yet proteolytically released domains can activate receptor-mediated signaling independent of force transmission (Swindle et al, 2001; Chiquet-Ehrismann & Chiquet, 2003; Iyer et al, 2008). Latent TGFβ provides another well-established example in which ECM deposition restricts ligand activity until integrin-dependent tension or proteolysis enables activation (Munger et al, 1999; Hinz, 2015). In this context, collagen-bound ACLP may be proteolytically processed contributing to spatially restricted and potentially sustained mechanical signaling within fibrotic niches, whereas soluble ACLP may exert broader biochemical effects through receptor-mediated pathways.

The relative contributions of ACLP presentation state to fibrogenic activation remain undefined and may vary across tissues and stages of disease. ACLP is a secreted protein that is detectable in circulation (Addona et al, 2011; Tao et al, 2021; Kattih et al, 2023) and also colocalizes within collagen-rich stromal–vascular niches (Jager et al, 2018). In the present study, we leveraged soluble versus collagen-bound presentation of ACLP as an experimental framework to isolate matrix-associated effects from canonical TGFβR1-driven signaling. Together with prior work demonstrating soluble ACLP signaling through TGFβ-associated pathways (Tumelty et al, 2014) and the matrix-dependent mechanical signaling described

here, these findings raise the possibility that ACLP exists as both a matrix-associated pool capable of supporting localized mechanical signaling and a soluble factor that exerts broader biochemical effects in vivo. Importantly, these findings do not imply that biochemical and mechanical signaling modes are mutually exclusive; rather, ACLP likely participates in a continuum of mechanical and cytokine signaling contexts that converge on overlapping transcriptional outputs.

### Broader implications and mechanistic significance

Together, our results define a collagen-bound ACLP pathway in which β1 integrin activation, RhoA/Rac1 signaling, and MRTFA nuclear translocation converge to drive early fibrogenic activation. This pathway operates independent of canonical TGFβR1 signaling and establishes ACLP as an ECM-derived mechanical signal rather than solely a soluble ligand. These findings broaden the mechanistic understanding of how ECM composition regulates stromal cell behavior and highlight ACLP as a structural mediator capable of coupling collagen organization to fibrogenic transcriptional programs. In adipose tissue, where stromal progenitor fate decisions determine the balance between adipogenesis and fibrosis, ECM-associated regulators such as ACLP may play a critical role in directing pathological tissue remodeling. Given the limitations of chronic TGFβ inhibition in clinical settings, targeting matrix-associated regulators such as ACLP may offer alternative strategies to modulate fibrotic tissue remodeling.

# Materials and Methods

### Generation of a recombinant protein

Generation of a recombinant ACLP was previously described (Tumelty et al, 2014). In brief, AD293 cells were stably transfected with plasmids encoding the BM40 signal peptide (Sparc), mouse sequence (26-1128; ACLP), and a C-terminal myc-His tag for detection and purification. These cells were cultured in suspension under serum-free conditions (HyClone SFM4HEK293; Cytiva). The conditioned medium was collected every 2 d for up to 2 wk and clarified (3,000g, 15 min). The supernatant was dialyzed against a potassium phosphate buffer (300 mM KCl, 3 mM $KH_2PO_4$, 7 mM $K_2HPO_4$). Protein was purified using an EconoFit Profinity IMAC (12009300; Bio-Rad) in a BioLogic Duo-Flow chromatography system (Bio-Rad). The eluted protein was concentrated with an Amicon centrifugal filter (UFC9010; Millipore) and dialyzed against PBS containing calcium and magnesium using 100,000 MWCO Float-A-Lyzer G2 Dialysis (G235035; Spectrum). Protein purity was determined by SDS–PAGE followed by SimplyBlue staining (LC6060; Invitrogen) according to the manufacturer's instructions.

### Cell stimulation by the ECM assay

Easy Coat 12 kPa Matrigen plates (SW6-EC; Matrigen) were coated with 0.05 mg/ml type I collagen (PureCol 5005; Advanced BioMatrix) in PBS (1.6 µg/cm²) polymerized in the presence or absence of

30 nM (3.75 $\mu$g/ml) recombinant ACLP for 2 h at 37°C, and washed with PBS. C3H/10T1/2 (CCL-226; ATCC) mesenchymal progenitors were maintained in DMEM supplemented with 10% FBS (SH30109; Cytiva) and 1% penicillin, streptomycin, and glutamine (PSG) (10378016; Gibco). Overnight serum-starved cells (1:1 DMEM:F12, 0.5% FBS, 1% PSG) were detached with 0.25% trypsin–EDTA for 4 min at 37°C, neutralized with DMEM, 10% FBS, and 1% PSG, then washed with serum-free DMEM, and collected by centrifugation (300$g$, 5 min) and resuspended. The washed cells were then maintained in suspension at a concentration of 3 × 10$^5$/ml with serum-free DMEM for 1 h at 37°C, then seeded onto coated hydrogels in low serum (1:1 DMEM:F12, 0.5% FBS, 1% PSG) at 30,000 cell/cm$^2$. Time 0 min corresponds to an aliquot of cells from suspension. At noted time points, stimulation was stopped by washing with cold PBS and either collected for protein analysis or fixed in 4% PFA (SC-281692; ChemCruz). For experiments with SB431542 (1614; R&D), 5 $\mu$M was added during cell seeding. For immunofluorescence analysis, 12 kPa polyacrylamide hydrogels (40% acrylamide: 2% bisacrylamide 1:0.45) were prepared and cast on 18-mm glass coverslips as described previously (Wong et al, 2003). After polymerization, the hydrogels were incubated in 2 mg/ml sterile dopamine hydrochloride solution in 50 mM Hepes (pH 8.5) for 15 min to coat the gel surface before being coated with collagen or ACLP as described above.

### ELISA-based gel incorporation assay

Collagen-coated 12 kPa polyacrylamide hydrogels were prepared and cast on 5-mm glass coverslips in the presence or absence of 30 nM (3.75 $\mu$g/ml) recombinant ACLP for 2 h at 37°C as described above and placed into a 96-well plate (12565501; Fisherbrand). Wells were washed with PBST, then blocked with PBS containing 4% BSA, washed again, then incubated in anti-Myc antibody (2276S, 1:1,000; Cell Signaling) diluted in PBS, 4% BSA for 1 h at RT. After three washes, the plates were incubated in HRP-conjugated secondary antibodies for 30 min at RT. After three washes, 50 $\mu$l TMB substrate was added, covered in tin foil, and incubated for 10 min at RT. 50 $\mu$l 1 M HCl was added, and the plate was read at 450 nM in the BioTek Synergy HT plate reader. Gelatin and 30 nM soluble ACLP were used as negative controls.

### Collagen polymerization assay

All reagents, tubes, and 96-well plates were precooled at 4°C to prevent premature collagen polymerization. Type I rat tail collagen (0.6 mg/ml) (354236; Corning) was diluted on ice with PBS containing 20 $\mu$g/ml ACLP or corresponding vehicle control. Solutions were mixed gently, dispensed at 100 $\mu$l per well into a clear, flat-bottom 96-well plate, and immediately transferred to a pre-warmed BioTek Synergy HT plate reader maintained at 37°C to initiate polymerization. Collagen polymerization was monitored by measuring absorbance as an index of light scattering (turbidity) from assembling collagen fibrils. Measurements were taken at 410 nm every 30 s for 45 min using BioTek Gen5 software (v3.11).

### Adhesion assays

A 96-well plate (12565501; Fisherbrand) was coated with 0.05 mg/ml type I collagen (PureCol 5005; Advanced BioMatrix) in PBS (1.6 $\mu$g/cm$^2$) polymerized in the presence or absence of 30 nM (3.75 $\mu$g/ml) recombinant ACLP for 2 h at 37°C. Wells were blocked with 1% BSA in serum-free DMEM for 1 min, washed, and seeded with 30,000 cells per well in 0.25% BSA in serum-free DMEM. Cells were allowed to adhere for 15–90 min at 37°C, then gently washed, and fixed in 1% glutaraldehyde for 5 min. Fixed cells were stained with 0.1% crystal violet for 30 min, extensively rinsed with water, air-dried, and solubilized in 0.2% Triton X-100 in water. Absorbance was read at 550 nm using a BioTek Synergy HT plate reader and BioTek Gen5 software (v3.11).

### Immunofluorescence

Cells were seeded on type I collagen–coated 18-mm coverslips. After 30, 60, 90 min, or 48 h, cells were fixed with 4% PFA (SC-281692; ChemCruz) for 30 min, permeabilized with PBS containing 0.1% Triton X-100 for 5 min, then blocked with PBS containing 5% goat serum, 1% BSA, and 0.1% Triton X-100 for 45 min. The cells were then incubated in primary antibody, diluted in PBS containing 1% goat serum, 1% BSA, and 0.1% Triton X-100, overnight at 4°C, then washed three times in PBS followed by a 1-h incubation in fluorescent secondary antibody at RT. After three washes in PBS, the coverslips were allowed to dry for 5 min, then mounted using ProLong Diamond Antifade with DAPI (P36966; Life Technologies). The following antibodies and stains were used: HUTS4 active $\beta$1 integrin (MAB2079Z, 1:100; Sigma-Aldrich), MRTFA (77098S, 1:100; Cell Signaling), pFAK (44-626, 1:100; BioSource), paxillin (05-417, 1:100; Upstate), phalloidin (A12381, 5 $\mu$l/200 $\mu$l; Invitrogen), goat anti-mouse IgG Alexa Fluor 488 (A-11001, 1:300; Invitrogen), and goat anti-rabbit IgG Alexa Fluor 488 (A-11008, 1:300; Invitrogen). Images were acquired (Axio Observer Z1; Carl Zeiss) using a 63× oil-immersion objective (NA 1.4) equipped with a digital camera (C10600/ORCA-R2; Hamamatsu Photonics).

### Cell image analysis

Images were analyzed in CellProfiler (v4.2.8) using five pipelines. For cell size measurements, phalloidin staining was segmented directly as the primary object using minimum cross-entropy thresholding with a diameter range of 100–100,000 pixels, and cell area was quantified and converted to $\mu m^2$. For paxillin, pFAK, and $\beta$1 integrin measurements, nuclei were first segmented using minimum cross-entropy thresholding and used to define single-cell boundaries from the respective fluorescence channels. Signal intensity was quantified on a per-cell basis using integrated intensity edge metrics. Quantification was performed on CellProfiler-defined single-cell objects. For MRTFA localization, nuclei were first segmented using minimum cross-entropy thresholding of the DAPI channel. Whole-cell boundaries were defined from phalloidin staining using minimum cross-entropy thresholding, and cytoplasmic regions were generated by subtracting nuclear masks from the phalloidin-defined cell masks. Mean MRTFA fluorescence intensity was then quantified separately within nuclear and

cytoplasmic compartments on a per-cell basis. CellProfiler pipelines used for these analyses are provided in Supplemental Data 1, Supplemental Data 2, Supplemental Data 3, Supplemental Data 4, and Supplemental Data 5.

### Rac1, RhoA, Cdc42 activation assays

GTPase activation was analyzed using G-LISA GTPase Activation Assay (BK135, Cytoskeleton) according to the manufacturer's instructions. In brief, $1 \times 10^6$ cells were cultured for 30 min per-cell stimulation by the ECM assay, then immediately lysed in 50 μl GL35/GL36 lysis buffer, clarified (10,000$g$ for 1 min), and snap-frozen in liquid nitrogen. Lysates were added to G-LISA well strips, incubated with antigen-presenting buffer, and primary and secondary antibodies, according to the manufacturer's protocol. HRP detection reagent was added for 5 min before HRP stop buffer was added for 10 min. Absorbance was read at 490 nm using a BioTek Synergy HT plate reader and BioTek Gen5 software (v3.11).

### G-actin/F-actin assay

G-actin-to-F-actin ratios were analyzed using G-actin/F-actin In Vivo Assay (BK037, Cytoskeleton) according to the manufacturer's instructions. In brief, $3 \times 10^5$ cells were cultured for 90 min per-cell stimulation by the ECM assay, then homogenized in F-actin stabilization buffer before being transferred to a prewarmed (37°C) ultracentrifuge rotor (F50L; FiberLite), and spun at 100,000$g$ for 1 h. The G-actin (supernatant) was collected, and the F-actin (pellet) was depolymerized in F-actin depolymerization buffer for 1 h on ice. The samples were diluted in 5× Laemmli sample buffer and subjected to SDS–PAGE and Western blotting for analysis.

### SDS–PAGE and Western blotting

Protein samples were boiled for 5 min at 95°C and run on a 4–12% or 12% Tris-glycine SDS–PAGE gels (XP04120, XP00120; Invitrogen). Samples were then transferred overnight onto 0.22-μm nitrocellulose membranes (10600094; Cytiva). Membranes were blocked with 4% milk in TBST for 45 min at RT and incubated with primary antibody in 4% milk TBST overnight at 4°C. Blots were incubated with HRP-conjugated secondary antibodies in 4% milk TBST for 1 h at RT. Chemiluminescent signal was detected using SuperSignal West Dura Substrate (34076; Thermo Fisher Scientific). Blots were imaged using ChemiDoc Imaging System (Bio-Rad). The following antibodies were used: anti-actin (AAN02-S; Cytoskeleton), anti-Myc antibody (2276S, 1:1,000; Cell Signaling), and mouse IgG HRP secondary (NA931, 1:4,000; Cytiva).

### Primary stromal progenitor isolation

All animal studies were approved by the Boston University Chobanian & Avedisian School of Medicine Institutional Animal Care and Use Committee. 12-wk-old, male and female C57BL/6 mice were euthanized; gonadal WAT depots were dissected under sterile conditions, washed 3× in cold PBS, and mechanically minced. Minced fat was digested in serum-free DMEM supplemented with 1% PSG (10378016; Gibco), 1% BSA (BP9706; Thermo Fisher

Scientific), and 1 mg/ml type I collagenase (LS004196; Worthington) for 45 min at 37°C on a nutator. The digestion was neutralized with DMEM supplemented with 10% FBS (SH30109; Cytiva) and 1% PSG. The cell suspension was filtered through a 100-μm cell strainer, then centrifuged at 300$g$ for 10 min. The cell pellet was resuspended in DMEM supplemented with 10% FBS and 1% PSG and plated for 48 h on hydrogels described in cell stimulation by the ECM assay.

### Real-time quantitative PCR (RT–qPCR)

Total RNA was isolated from cultured primary stromal cells and 10T1/2 cells using GeneJET RNA Purification Kit (K0731; Thermo Fisher Scientific) according to the manufacturer's instructions. cDNA was synthesized from 250–500 ng of total RNA using Luna-Script RT SuperMix Kit (E3010; NEB), then diluted 1:5 in nuclease-free water. qPCR was performed using Luna Universal qPCR Master Mix (M3003; NEB) with a CFX Opus Real-Time PCR system (Bio-Rad). Relative gene expression (fold change) was calculated using the ΔΔCT method using PPIA gene expression to normalize samples. All primer pairs were obtained from the MGH Primer Bank and are detailed in Table S4.

### Bulk RNA sequencing

Total RNA was isolated as described above. Library preparation and bulk RNA sequencing were performed by Novogene using Illumina NovaSeq X Plus Series (PE150) with 150-bp paired-end reads and a 20 million read depth. Clean reads were mapped to the mouse reference genome (GRCm39/mm39) using HISAT2 software (v2.1.0). Differential gene expression analysis was performed on raw counts using the R statistical package DESeq2 (v1.28.1). Genes were considered differentially expressed if they exhibited an adjusted $P$-value < 0.05 (Benjamini–Hochberg FDR correction) and an absolute $\log_2$FC > 0.58. Functional enrichment analyses were performed in R (v4.5.1) using the clusterProfiler (v4.16.0) package. Gene set enrichment analysis (GSEA) was performed against the KEGG database. Pathways with a false discovery rate (FDR q < 0.1) were considered significantly enriched. Per-gene Pearson correlation analysis was performed in R (v4.5.1) using $\log_2$FC values obtained from DESeq2 contrasts. Correlation coefficients were calculated using the cor() function (method = "pearson"), and results were visualized in ggplot2 with the y = x diagonal included as a reference. Genes were classified based on the difference in $\log_2$FC between conditions: independent ($|\Delta\log_2$FC$|$ < 0.3), attenuated ($\Delta\log_2$FC ≥ 0.3 and adjusted $P$ < 0.05), or dependent (adjusted $P$ ≥ 0.05).

### Statistical analysis

Statistical analyses were performed using GraphPad Prism version 5. Data are presented as the mean ± S.D. Unpaired two-tailed $t$ tests with Welch's correction were used for comparisons between two groups, whereas one-way ANOVA was applied for comparisons among multiple groups. Statistical significance was defined as $P$ < 0.05 unless noted.

# Data Availability

The RNA-sequencing data that support the findings of this study are publicly available on Sequencing Read Archive (SRA) under BioProject PRJNA1368594 and processed data in the Gene Expression Omnibus (GEO) repository under accession numbers GSE312027 (10T1/2 cells) and GSE324887 (primary stromal cells).

# Supplementary Information

# Acknowledgements

We would like to thank Dr. Mikel Garcia-Marcos and Dr. Arthur Marivin for providing experimental protocols and expertise, as well as Dr. Joe Tien and Dr. Alex Siebel for their feedback and guidance on optimizing the hydrogel system. This work was supported by the National Institutes of Health grant R01DK132080 (to MD Layne), R01DK134534 (to Stephen R Farmer and MD Layne), and F31DK139746 (to CL Frosti).

## Author Contributions

CL Frosti: conceptualization, data curation, formal analysis, funding acquisition, validation, investigation, visualization, methodology, and writing—original draft, review, and editing.
D Yeritsyan: data curation and writing—review and editing.
MD Layne: conceptualization, supervision, funding acquisition, investigation, project administration, and writing—review and editing.

## Conflict of Interest Statement

The authors declare that they have no conflict of interest.

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
