## [Reviewer comments · Life Science Alliance]

Aortic carboxypeptidase-like protein potentiates β 1 integrin signaling in mesenchymal progenitors

Cheyenne Frosti, Diana Yeritsyan, and Matthew Layne

DOI: <https://doi.org/10.26508/lsa.202503600>

Corresponding author(s): Matthew Layne, Boston University

Review Timeline:

Submission Date:	2025-12-15
Editorial Decision:	2026-01-30
Revision Received:	2026-02-20
Editorial Decision:	2026-03-05
Revision Received:	2026-03-18
Accepted:	2026-03-20

Scientific Editor: Sarita Hebbar

Transaction Report:

January 30, 2026

Re: Life Science Alliance manuscript #LSA-2025-03600-T

Dr. Matthew D Layne
Boston University
Biochemistry & Cell Biology
72 E. Concord St
Boston, MA 02118

Dear Dr. Layne,

Thank you for submitting your manuscript entitled "Aortic carboxypeptidase-like protein activates fibroblasts through β 1 integrin signaling" to Life Science Alliance.

The manuscript was assessed by three expert reviewers, whose comments are appended to this letter. As you will note, the reviewers are consistent in their positive assessment and value of this work. That said they have all raised some concerns that preclude publication at this stage.

We agree with Reviewers 1 and 3 that you must address the concern on the cell-type used in this study. We leave it to your choice to demonstrate the major findings in relevant cell types or primary fibroblasts as suggested by both reviewers. In the absence of this, you must modify your claims and references to examining fibroblasts everywhere in the manuscript text.

In terms of technical concerns, we concur with Reviewer 2 that you must provide evidence for purity of the ACLP-preparation. We also agree with this reviewer that you must provide additional experimental evidence for incorporation of ACLP into the collagen matrix. In the absence of additional evidence, you must moderate your conclusion and acknowledge the limitation of ELISA-assays/need for other evidence. Further we point you to the comments of Reviewer 2 on imaging methods and presentation of associated results.

Finally we agree with Reviewers 1 and 3 that some validation for RNASeq findings will strengthen your manuscript.

In line with the reviewers overall evaluation, we invite you to submit a revised manuscript addressing the reviewers' comments.

When submitting the revision, please include a letter addressing the reviewers' comments point by point. While a rebuttal must respond to all points in some form, additional experiments to resolve these points, other than indicated above, are not required.

I would be happy to discuss the revision in more detail via email or phone/videoconferencing. Please let me know which option you prefer, if any.

Thank you for this interesting contribution to Life Science Alliance. We hope that the comments below will prove constructive as your work progresses, and we are looking forward to receiving your revised manuscript.

Sincerely,

Sarita Hebbar, PhD
Scientific Editor
Life Science Alliance
<http://www.lsajournal.org>

B. MANUSCRIPT ORGANIZATION AND FORMATTING:

Reviewer #1 (Comments to the Authors (Required)):

The paper is reasonably straightforward and continues and extends prior work from this lab on MRTFA. However, the novelty relies on the effects of ALCP on 10T1/2 fibroblasts, but not on the underlying mechanism uncovered, which is arguably relatively well-established. That said, the paper should be a good addition to the literature.

1. It is perhaps surprising that the authors have focused on 10T1/2 cells which, although being fibroblast-like, are considered to be sarcomas. It is better for the investigators to use primary cells or at least commercially available fibroblast "lines". The authors should explain in more detail in the text why they selected these cells. Without using perhaps more relevant cells, it is difficult to claim eg in the title that they are examining "fibroblasts". It is reasonable to state in the title and elsewhere that C3H/10T1/2 cells are being investigated. The authors do state "10T1/2 fibroblasts" elsewhere in the text.
2. The actual data presented appear clear. However, I request that dimensions of the scale bars in Figs 2 and 4 be provided.
3. Sometimes the authors have relied on review articles rather than the primary literature. Prior art linking integrin beta 1 and rac1 activity-and MRTFA-- in fibroblasts to fibroblast activation/fibrosis could be provided in the discussion to support the conclusions drawn. Pertinent papers could be: PMID: 18576327 PMID: 19714619 PMID: 26763945 PMID: 25955164 PMID: 24706986 PMID: 19823586.
3. The sources of the reagents need to be added.
4. Further validation of the RNAseq data by following up on a few markers is suggested

Reviewer #2 (Comments to the Authors (Required)):

SUMMARY

The study by Frosti et al. makes a valuable contribution by showing that collagen-bound Aortic Carboxypeptidase-like Protein (ACLP) promotes fibroblast activation through an integrin-dependent RhoA/Rac1 signaling cascade that drives actin remodeling and nuclear translocation of Myocardin-related Transcription Factor A (MRTFA). Using imaging, biochemical assays and RNA-seq analysis, the authors show that this activation modality is predominantly independent of canonical TGFBR1 signaling, previously associated with soluble ACLP-mediated response. Overall, these findings define a matrix-associated mode of ACLP signaling in fibroblasts that drives pro-fibrotic and matrix-remodeling gene expression. The study is well designed, and the evidence supporting the main conclusions is solid. However, additional methodological validation would improve the rigor of key

experimental setups underlying their conclusions.

MAJOR POINTS

1. The main conclusions rely heavily on experiments using recombinant myc-tagged ACLP, yet its purity is not shown. To ensure the observed phenotypes can be attributed specifically to ACLP (i.e., matrix-ACLP), the authors should include a protein gel (i.e., SDS-PAGE) confirming the purity of this preparation.
2. (Fig. S1) The authors used ELISA to claim incorporation of ACLP into the collagen matrix. While this approach demonstrates collagen-associated retention of ACLP, ELISA alone is insufficient to support this claim. If the authors wish to claim that ACLP is indeed incorporated into collagen following co-polymerization, they should complement these data with biochemical fractionation (e.g., Western blot analysis of collagen-bound versus released fractions following collagenase treatment). Alternatively, the authors could rephrase their interpretation and acknowledge the limitations of ELISA-based assays.

MINOR POINTS

3. (Fig S1A) The figure legend does not fully describe the experimental conditions shown, particularly the use of gelatin as a control. For example, it is unclear whether "matrix-ACLP" refers to Col1/ACLP, Gelatin/ACLP or both. To improve clarity, the authors should describe the controls used and the rationale for their inclusion.
4. (Fig 4A) The panel showing actin networks under different conditions is too small to allow visual comparison of structural details and does not appear to be gain-matched across conditions (e.g., DAPI signal is markedly lower in the first panel). To facilitate fair comparison, the authors should enlarge the images or add zoomed insets and ensure that image acquisition and intensity scaling are matched across all conditions.
5. While the discussion addresses the mechanical aspects of ACLP signaling, it could be strengthened by expanding on the physiological implications of these findings. In particular, it would be interesting if the authors could discuss the potential in vivo relevance of having two distinct modes of pathway activation leading to similar downstream responses. For example, could collagen-bound ACLP contribute to more spatially restricted or long-lasting signaling compared with its soluble form?
6. The methods section lacks some important details. To facilitate reproducibility of the reported results, the authors should include complete information on vendors and catalog numbers for the anti-Myc and anti-MRTFA antibodies, as well as for SB431542; and provide a more complete description of ELISA assay procedures.

Reviewer #3 (Comments to the Authors (Required)):

In this manuscript by Frosti and colleagues, the authors present data evaluating the effect of aortic carboxypeptidase-like protein (ACLP) on fibroblast phenotype and mechanosensing. While data are compelling in suggesting a means to modulate fibroblast responses, some limitations need to be addressed.

1. The most critical issue concerns the fibroblast source used in this work. The cell line used in this study is derived from mouse embryo. It has been shown that fibroblasts are highly heterogeneous, with phenotypes that vary by organ, age, sex, and species. While the embryonic cell line responds robustly to ACLP, the same needs to be demonstrated for primary fibroblasts.
2. The images should be accompanied by Western Blots quantifications (Fig. 2 and Fig.4).
3. Please include how many cells were analyzed in Fig. 4.
4. Please validate RNA seq data by choosing some relevant targets that support this study.

Point by Point Response to Reviewer's Comments

We thank the reviewers for their thoughtful and constructive evaluation of this work. The strengths highlighted that the “study is well designed and that the evidence supporting the main conclusions is solid” and that the “data are compelling in suggesting a means to modulate fibroblast responses.” However, based on the reviewer's feedback, we have carefully revised the manuscript to include justification of the use of 10T1/2 fibroblasts, incorporated primary adipose-derived cell data, validated the recombinant ACLP matrix model, and strengthened scientific rigor throughout the methods. We thank the reviewers for their thoughtful feedback, which we believe has significantly improved the manuscript and strengthened it for publication. Specific revisions are outlined below, where the **reviewer comments** are in **bold** and our responses are in plain text.

Reviewer #1

1. It is perhaps surprising that the authors have focused on 10T1/2 cells which, although being fibroblast-like, are considered to be sarcomas. It is better for the investigators to use primary cells or at least commercially available fibroblast "lines". The authors should explain in more detail in the text why they selected these cells. Without using perhaps more relevant cells, it is difficult to claim eg in the title that they are examining "fibroblasts". It is reasonable to state in the title and elsewhere that C3H/10T1/2 cells are being investigated. The authors do state "10T1/2 fibroblasts" elsewhere in the text.

We expanded the rationale for using 10T1/2 cells as a fibroblast-like mesenchymal progenitor model and added appropriate references describing their multipotency and fibroblast-like properties (Putra et al., 2023; Tang et al., 2004; Pinney and Emerson, 1989). Although 10T1/2 cells can give rise to sarcomas following induction with chemical carcinogens or radiation, they are otherwise considered non-tumorigenic (Kirschmeier et al., 1982). We selected this cell line because of its fibroblast-like, multipotent nature and because our laboratory has previously demonstrated that 10T1/2 cells are a robust and sensitive model for studying ACLP-mediated responses (Jager et al., 2018). We have also expanded the manuscript to include primary adipose derived cells as one established role for ACLP is modifying differentiation pathways. 10T1/2 cells also serve as a model for adipose progenitor differentiations.

Please refer to the revised second paragraph of the Results section:

“The following experiments were performed using 10T1/2 cells, a well-established mouse mesenchymal progenitor cell line that exhibits fibroblast-like morphology, ECM production, and robust responsiveness to pro-fibrotic signaling, making it a useful *in vitro* model for studying fibroblast activation pathways (Pinney and Emerson, 1989; Tang et al., 2004; Putra et al., 2023).”

2. The actual data presented appear clear. However, I request that dimensions of the scale bars in Figs 2 and 4 be provided.

Thank you for bringing this to our attention. Scale bar size was already included in figure legends for Fig. 2 and 4 and are now also displayed directly within the immunofluorescence images for added clarity.

3. Sometimes the authors have relied on review articles rather than the primary literature. Prior art linking integrin beta 1 and rac1 activity-and MRTFA-- in fibroblasts to fibroblast activation/fibrosis could be provided in the discussion to support the conclusions drawn.

We incorporated additional primary references supporting integrin signaling, mechanotransduction, and fibroblast activation.

Please refer to the revised Results sections:

“ β 1 integrins are established collagen-binding receptors in fibroblasts (Hynes, 2002; Sun et al., 2016), and their transition from an inactive conformation to an active conformation is a critical step in transmitting ECM cues into intracellular adhesion and signaling complexes (Campbell and Humphries, 2011; Liu et al., 2009).”

“These GTPases serve as critical effectors that link ECM cues to cytoskeleton remodeling and fibroblast behavior (Hall, 2012; Liu et al., 2008; Xu et al., 2009).”

“Actin polymerization regulates gene expression through several mechanisms including the release of MRTFA from G-actin, enabling its nuclear translocation and co-activation of SRF-dependent transcription (Small, 2012; Haak et al., 2014).”

“This actin-MRTFA axis is a well-established route by which matrix stiffness and integrin signaling influence fibroblast gene expression (Yang and Plotnikov, 2021; Small, 2012; Fearing et al., 2019; Shiwen et al., 2015).”

3. The sources of the reagents need to be added.

Thank you for bringing this to our attention. We have gone through the Methods and any missing reagent sources have been added.

4. Further validation of the RNAseq data by following up on a few markers is suggested

To validate transcriptional changes identified by RNA sequencing, we performed qPCR analysis of representative differentially expressed genes (**new Fig. S3**). These results confirm the directionality and magnitude of key gene expression changes induced by matrix-bound ACLP.

Please refer to the revised Results sections:

“To validate the findings, we quantified the expression of representative differentially expressed genes (DEG) between the three conditions and observed consistency with the RNA-seq (**Fig. S3**).”

Reviewer #2

1. The main conclusions rely heavily on experiments using recombinant myc-tagged ACLP, yet its purity is not shown. To ensure the observed phenotypes can be attributed specifically to ACLP (i.e., matrix-ACLP), the authors should include a protein gel (i.e., SDS-PAGE) confirming the purity of this preparation.

We added multiple, complementary validation approaches for recombinant ACLP (**new Fig. S1**), including SDS-PAGE and Western blot analysis and activity in established collagen polymerization assays to verify biological activity.

Please refer to the revised Results sections:

“To study early fibroblast responses to ACLP, we generated recombinant ACLP from mammalian cells (Tumelty et al., 2014), verified its purity by SDS-PAGE (**Fig. S1A**), and confirmed identity by immunoblotting using a myc-tag antibody (**Fig. S1B**). Using a previously established cell-free collagen

polymerization assay, ACLP increased collagen fibrillogenesis (**Fig. S1C**), consistent with prior reports and validating its functional activity (Blackburn et al., 2018).”

2. The authors used ELISA to claim incorporation of ACLP into the collagen matrix. While this approach demonstrates collagen-associated retention of ACLP, ELISA alone is insufficient to support this claim. If the authors wish to claim that ACLP is indeed incorporated into collagen following co-polymerization, they should complement these data with biochemical fractionation (e.g., Western blot analysis of collagen-bound versus released fractions following collagenase treatment). Alternatively, the authors could rephrase their interpretation and acknowledge the limitations of ELISA-based assays.

3. (Fig S1A) The figure legend does not fully describe the experimental conditions shown, particularly the use of gelatin as a control. For example, it is unclear whether "matrix-ACLP" refers to Col1/ACLP, Gelatin/ACLP or both. To improve clarity, the authors should describe the controls used and the rationale for their inclusion.

We agree that ELISA-based assays alone cannot definitively demonstrate physical incorporation of ACLP into collagen matrices. We have therefore revised the text to clarify that ELISA demonstrates collagen-associated retention of ACLP, and to reference complementary biochemical and structural evidence supporting matrix incorporation, as well as to describe the controls used to support collagen association. Specifically, prior studies from our lab have detected ACLP within the ECM fraction of collagen matrices (Schissel et al., 2009) and direct incorporation of ACLP into engineered collagen fibers (Vishwanath et al., 2020). In our assay, ACLP was retained only when present during collagen polymerization and not when added in soluble form after gel formation.

Please refer to the revised Results section:

“Collagen-associated retention of ACLP following co-polymerization was assessed by an ELISA-based assay (**Fig. S1D**). While this approach does not directly demonstrate physical incorporation, ACLP was retained only when present during collagen polymerization and not when added in soluble form after gel formation. Furthermore, ACLP was not retained on gelatin substrates, indicating collagen-specific matrix association. These findings are consistent with prior studies from our lab demonstrating ACLP incorporation into engineered collagen fibers (Vishwanath et al., 2020) and detection of ACLP within the ECM fraction of collagen matrices (Schissel et al., 2009).”

Furthermore, the figure and figure legend (Fig. S1) have been revised to clarify experimental conditions controls.

Revised Fig S1 legend:

“Incorporation of recombinant myc-tagged ACLP into collagen I coated hydrogels was confirmed using an ELISA-based detection assay. Absorbance was measured at 450 nm. Gelatin substrate was used as a negative control to assess collagen specificity. Matrix-ACLP refers to ACLP incorporated into the substrate during polymerization whereas, soluble ACLP was added in solution after substrate polymerization.”

4. (Fig 4A) The panel showing actin networks under different conditions is too small to allow visual comparison of structural details and does not appear to be gain-matched across conditions (e.g., DAPI signal is markedly lower in the first panel). To facilitate fair comparison, the authors should enlarge the images or add zoomed insets and ensure that image acquisition and intensity scaling are matched across all conditions.

All images in Fig. 4 were acquired using identical microscope settings (lamp power, detector gain, and exposure) and were subjected to the same linear intensity scaling during image processing. Phalloidin labels filamentous actin rather than total actin; therefore, conditions with fewer or thinner stress fibers exhibit lower overall fluorescence intensity. Minor variations in apparent signal intensity in DAPI for example, reflect local differences in gel surface flatness and focal plane rather than differences in acquisition parameters. We have now enlarged the actin panels and added insets to facilitate visualization of cytoskeletal architecture.

5. While the discussion addresses the mechanical aspects of ACLP signaling, it could be strengthened by expanding on the physiological implications of these findings. In particular, it would be interesting if the authors could discuss the potential *in vivo* relevance of having two distinct modes of pathway activation leading to similar downstream responses.

The Discussion has been expanded to emphasize the significance of identifying two distinct modes of ACLP-dependent pathway activation that converge on pro-fibrotic gene expression and may cooperatively contribute to fibrotic remodeling *in vivo*.

Please refer to the Discussion section “Presentation-state dependent signaling of ACLP in fibrotic microenvironments”:

“Collagen-bound ACLP promoted fibroblast spreading, whereas soluble ACLP did not increase cell spreading on collagen matrices, indicating that matrix association is required for ACLP to support force-dependent signaling. Similar presentation-dependent regulation has been described for other matricellular proteins. Tenascin-C modulates integrin engagement and mechanotransduction when embedded within fibronectin-rich matrices, yet proteolytically released domains can activate receptor-mediated signaling independent of force transmission (Chiquet-Ehrismann et al., 2003; Swindle et al., 2001; Iyer et al., 2008). Latent TGF β provides another well-established example in which ECM deposition restricts ligand activity until integrin-dependent tension or proteolysis enables activation (Munger et al., 1999; Hinz et al., 2015). In this context, collagen-bound ACLP may be proteolytically processed contributing to spatially restricted and potentially sustained mechanical signaling within fibrotic niches, while soluble ACLP may exert broader biochemical effects through receptor-mediated pathways.

The relative contributions of ACLP presentation state to fibroblast activation remain undefined and may vary across tissues and stages of disease. ACLP is a secreted protein that is detectable in circulation (Kattih et al., 2023; Addona et al., 2011; Tao et al., 2021) and also co-localizes within collagen-rich stromal-vascular niches (Jager et al., 2018). In the present study, we leveraged soluble versus collagen-bound presentation of ACLP as an experimental framework to isolate matrix-associated effects from canonical TGF β R1-driven signaling. Together with prior work demonstrating soluble ACLP signaling through TGF β -associated pathways (Tumelty et al., 2014) and the matrix-dependent mechanical signaling described here, these findings raise the possibility that ACLP exists as both a matrix-associated pool capable of supporting localized mechanical signaling and a soluble factor that exerts broader biochemical effects *in vivo*. Importantly, these findings do not imply that biochemical and mechanical signaling modes are mutually exclusive; rather, ACLP likely participates in a continuum of mechanical and cytokine signaling contexts that converge on overlapping transcriptional outputs.”

6. The methods section lacks some important details. To facilitate reproducibility of the reported results, the authors should include complete information on vendors and catalog numbers for the anti-Myc and anti-MRTFA antibodies, as well as for SB431542; and provide a more complete description of ELISA assay procedures.

Thank you for bringing this to our attention, missing reagent sources have been added, and additional experimental details for Fig. S1 are now included in the Methods.

Reviewer #3

1. While the embryonic cell line responds robustly to ACLP, the same needs to be demonstrated for primary fibroblasts.

We thank the reviewer for comment and appreciate the opportunity to address since Reviewer #1 similarly raised this point. In addition, we now included experiments using primary mouse adipose stromal vascular fraction (SVF) stromal cells demonstrating conserved responses to matrix-bound ACLP (**new Fig. 6**), supporting the physiological relevance of our findings.

Please refer to the Results section “Collagen-bound ACLP promotes mechanosensitive transcriptional programs in primary SVF stromal cells”:

Given prior evidence implicating ACLP in adipose tissue fibrosis and stromal remodeling *in vivo* (Jager et al., 2018) and the capacity of 10T1/2 to commit to an adipogenic lineage (Tang et al., 2004), we next assessed whether collagen-bound ACLP elicits similar transcriptional responses in primary stromal cells. Stromal vascular fraction (SVF) cells isolated from mouse epididymal white adipose tissue (eWAT) were plated on 12 kPa polyacrylamide hydrogels coated with type I collagen alone (col1) or collagen polymerized in the presence of ACLP (col1-ACLP) and cultured for 48 hours (**Fig. 6A**).”

2. The images should be accompanied by Western Blots quantifications (Fig. 2 and Fig.4).

We agree that Western blot quantification is valuable for assessing changes in bulk protein abundance. However, the primary readouts in Figs 2 and 4 reflect spatially localized and conformation-dependent events including focal adhesion clustering, integrin activation, and nuclear localization of transcriptional regulators, that are not well captured by bulk lysate approaches. Accordingly, we quantified these phenotypes using immunofluorescence with large biological replicate numbers (>100 cells per condition) and report single-cell quantitative metrics using CellProfiler-based image analysis of focal adhesion parameters, integrin activation, and nuclear-to-cytoplasmic localization. We have clarified this rationale in the Results section to emphasize that imaging-based quantification was selected to capture spatial resolution.

Please refer to the Results section:

“Because the signaling events examined here, including integrin conformational activation and focal adhesion clustering, depend on protein localization and conformational state rather than total protein abundance, we employed quantitative, single-cell immunofluorescence approaches rather than bulk immunoblotting to assess pathway activation.”

3. Please include how many cells were analyzed in Fig. 4.

The number of cells analyzed have been added to the legends for Fig. 3 and 4. Fig 3.

Please refer to the figure legends:

“Fig. 3. Collagen-bound ACLP enhanced RhoA and Rac1 GTPase activity. 10T1/2 fibroblasts were seeded ... was measured using G-LISA activation assays (Cytoskeleton, Inc.). Absorbance was measured at 490 nm. Data represent mean \pm SD from three independent experiments (3×10^5 cells per n; n=3).”

“Fig. 4. Collagen-bound ACLP increases F-actin assembly and MRTFA nuclear accumulation. 10T1/2 fibroblasts were seeded ... Quantification of the G-actin/F-actin ratio from three independent experiments (3×10^5 cells per n; n=3) ... (D) Quantification of MRTFA localization. Data represent mean \pm SD from three independent experiments. At least 30 cells per condition per replicate were analyzed (≥ 100 cells total).”

4. Please validate RNA seq data by choosing some relevant targets that support this study.

As similarly requested by Reviewer #1, we validated transcriptional changes identified by RNA sequencing, we performed qPCR analysis of representative differentially expressed genes (**new Fig. S3**). These results confirm the directionality and magnitude of key gene expression changes induced by matrix-bound ACLP.

Please refer to the revised Results sections:

“To validate the findings, we quantified the expression of representative differentially expressed genes (DEG) between the three conditions and observed consistency with the RNA-seq (**Fig. S3**).”

March 5, 2026

RE: Life Science Alliance Manuscript #LSA-2025-03600-TR

Dr. Matthew D Layne
Boston University
Biochemistry & Cell Biology
72 E. Concord St
Boston, MA 02118

Dear Dr. Layne,

Thank you for submitting your revised manuscript entitled "Aortic carboxypeptidase-like protein activates fibroblasts through β 1 integrin signaling". Your revised manuscript was assessed by all the original reviewers. Reviewer 2 notes that their original comments were addressed. However the two other reviewers point to major concerns that have not been addressed. Reviewers 1 and 3 note the continued use of "fibroblasts", in your description of this work, and Reviewer 3 notes the absence of quantification of Western Blots.

We agree with Reviewers 1 and 3, and in line with our previous decision letter, that you must refrain from referring to your model system (10T1/2) as fibroblasts. In line with this, please remove all references to fibroblasts from your title/description/conclusion, and replace with an accurate description or follow the suggestion of Reviewer 1. With the comments from Reviewer 2 on quantification of Western Blotting, whilst we appreciate that importance of their suggestion, we also noted your response about spatial differences and the rationale to quantify immunostaining data, and do not recommend further change.

In line with the reviewers' overall evaluation, we would be happy to publish your paper in Life Science Alliance pending resolution of the above points and final revisions necessary to meet our formatting guidelines. We request you to submit a revised manuscript document with all these changes highlighted along with a point-by-point response to the reviewer's comments.

MANUSCRIPT ORGANIZATION AND FORMATTING:

To avoid unnecessary delays in the acceptance and publication of your paper, please read the following information carefully. Full guidelines are available on our Instructions for Authors page, <https://www.life-science-alliance.org/authors>

- Please include a section in the methods describing the isolation of primary stromal cells from mice. This section must also include a statement confirming that all experiments were performed in accordance with relevant guidelines and regulations and identifying the institutional and/or licensing committee approving the experiments as listed in LSA' guidelines (<https://www.life-science-alliance.org/editorial-policies#animals>).
- Please expand on the recombination and purification of ACLP.
- Please include a supplementary file with the code from your CellProfiler pipeline for all quantification presented in this work.
- Please expand your methods to include RNA extraction and use in RNA-seq for these primary stromal cells.
- Please provide a scale bar for zoomed-in images shown in Figure 2A-C and 4A.
- Please include source information for the anti-MRTFA antibody (also pointed out by Reviewer 2) or its preparation if not obtained from commercial sources.
- Thank you for providing a statement on data availability. Please explicitly state if RNA Seq data from 10T1/2 and primary stromal cells are associated with the same accession ID. If not, please provide the corresponding accession ID.
- Please add the X and Bluesky handles of your host institute/organization, as well as your own, and/or one of the authors, in our system.
- Please upload your Tables in editable .doc or Excel format.
- Please rename "Competing interests" to "Conflict of Interest."
- Please use the [10 author names et al.] format in your references (i.e., limit the author names to the first 10).
- Please add an Author Contributions section to your main manuscript text.
- Please be sure that the authorship listing and order is correct

We welcome submissions of potential cover images for the issue of LSA in which your work would appear. If you have high quality images associated with this work, please feel free to email these, with a caption, to the journal office.

LSA encourages authors to provide a 30-60 second video where the study is briefly explained. We will use these videos on social media to promote the published paper and the presenting author (for examples, see <https://docs.google.com/document/d/1-UWCfbE4pGcDdcgzcmiuJl2XMBJnxKYeqRvLLrLSo8s/edit?usp=sharing>). Corresponding or first-authors are welcome to submit the video. Please submit only one video per manuscript. The video can be emailed to contact@life-science-alliance.org

FINAL FILES:

The following items are required for acceptance.

The license to publish form must be signed before your manuscript can be sent to production. A link to the license to publish form will be available to the corresponding author only. Please take a moment to check your funder requirements.

Thank you for your attention to these final processing requirements. Please revise and format the manuscript and upload materials as soon as you are able.

Thank you for this interesting contribution to the literature. We look forward to publishing your paper in Life Science Alliance.

Sincerely,

Sarita Hebbar, PhD
Scientific Editor
Life Science Alliance
<http://www.lsjournal.org>

Reviewer #1 (Comments to the Authors (Required)):

The authors have clarified the choice of cell types.

I appreciate their updated description of 10t1/2 cells as well, as their use of primary fat-derived mesenchymal progenitor cells.

However the authors still have not used primary fibroblasts.

They have used models of mesenchymal progenitor cells.

Although the new data and presentation are interesting, as a consequence of these changes mentions of "fibroblasts" need to

be expunged from the paper and replaced with "mesenchymal progenitor cells" as no "fibroblasts" are used in this report.

Alternatively, they should use primary fibroblasts.

At very least, they have to justify why they have used this new cell type (adipogenic precursors or mesenchymal progenitors)

Reviewer #2 (Comments to the Authors (Required)):

The authors have thoroughly addressed the two major points I raised by including supporting data that confirms the purity and activity of ACLP and by moderating their claim regarding the Col1-associated state of ACLP based on the available ELISA data. They have also satisfactorily addressed my minor comments. As a final minor detail, please include the source information for the anti-MRTFA antibody, whether commercial or homemade. Overall, the manuscript has substantially improved and provides convincing evidence that collagen-associated ACLP drives TGFBR1-independent pro-fibrotic fibroblast activation.

Reviewer #3 (Comments to the Authors (Required)):

In a previous manuscript review, authors were asked to include primary fibroblasts to establish that, similarly to the embryonic cell line, they respond to ACLP. Instead, authors included results from adipose stromal vascular fraction cells, which did not improve this manuscript.

Another comment that was not addressed concerns the quantification of proteins in Figs. 2 and 4. The presented images are not convincing (especially pFAK), therefore, the pFAK/FAK ratio and MRTFA nuclear translocation should be assayed by Western Blot.

Point by Point Response to Reviewer's Comments

We thank the reviewers for their thoughtful and constructive evaluation of this work. In response to the reviewers' concerns regarding the identity of the cell models used, we have carefully revised the manuscript to clarify that the 10T1/2 cell line represents **mesenchymal progenitor cells** rather than primary fibroblasts. We also thank the editor for their helpful feedback and have addressed all requested formatting and content revisions including submitting the copyright. Specific responses to reviewer comments are provided below. Reviewer comments are shown in **bold**, followed by our responses in plain text.

Editor Comments

We thank the editor for these helpful comments. All requested revisions have been addressed in the revised manuscript and associated files:

Content and Methods Updates

- The RNA-seq data availability statement has been clarified to specify the accession IDs for datasets generated from 10T1/2 and primary stromal cells.
 - 10T1/2 GEO accession number GSE312027
 - SVF GEO accession number GSE324887
- "Materials and methods" have been expanded to include details on "Primary Stromal Progenitor Isolation", including a statement confirming that all animal experiments were performed in accordance with relevant guidelines and regulations and approved by the appropriate institutional committee.
- Additional methodological details describing RNA extraction and preparation for RNA-seq from primary stromal progenitors have been included.
- "Materials and methods" have been expanded to provide details on "Generation of recombinant protein"

File Updates

- The 5 CellProfiler pipelines used for image quantification cannot be uploaded into the review system. We can provide these by email since they are .cproj files
- Tables have been uploaded in editable format (Excel).

Image Updates

- Scale bars have been added to the zoomed-in images in Figures 2A-C
- The representative image in Figure 4A and zoomed in scale bars have also been updated.

Editorial and Formatting Changes

- References have been reformatted to the [10 authors et al.] format.
- An Author Contributions section has been added to the first page of the main manuscript text.
- "Competing interests" has been renamed to "Conflict of Interest".
- The authorship listing and order have been reviewed and confirmed as correct.
- X and Bluesky handles for the host institute have been added in the submission system as requested.

Reviewer #1

1. However the authors still have not used primary fibroblasts. They have used models of mesenchymal progenitor cells. Although the new data and presentation are interesting, as a consequence of these changes mentions of "fibroblasts" need to be expunged from the paper and replaced with "mesenchymal progenitor cells" as no "fibroblasts" are used in this report.

We thank the Reviewers for emphasizing the importance of clearly defining the cellular models used in this study. To address this comment and improve clarity, we have revised the manuscript to consistently describe the 10T1/2 cell line as mesenchymal progenitor cells, which can adopt fibroblast-like phenotypes or fibrogenic activation under fibrogenic stimuli. Accordingly, terminology throughout the manuscript has been revised to avoid implying that fibroblasts were used. Importantly, these terminology updates do not alter the experimental findings but clarify the cellular context in which ACLP signaling was examined.

These changes have been implemented throughout the manuscript, including:

- Title, which now reflects fibrogenic activation of mesenchymal progenitors.
- Introduction, where we clarify that adipose stromal vascular fraction contains stromal and mesenchymal progenitor populations capable of fibrogenic differentiation.
- Methods, where the 10T1/2 cell line is explicitly defined as a mesenchymal progenitor.
- Results, where descriptions of cell behavior have been revised to refer to mesenchymal progenitor cells or stromal cells rather than fibroblasts.
- Discussion, where interpretations have been updated to emphasize fibrogenic activation and stromal progenitor cell behavior.

For example, the Results section now states:

“Using 10T1/2 mouse mesenchymal progenitor cells, we identified a mechanism by which collagen-bound ACLP induces fibrogenic activation through β 1 integrin-mediated signaling.”

These revisions ensure that the terminology used throughout the manuscript accurately reflect the cellular models employed while preserving the mechanistic conclusions of the study.

Reviewer #2

1. As a final minor detail, please include the source information for the anti-MRTFA antibody, whether commercial or homemade.

We thank Reviewer #2 for noting this oversight. The source information for the anti-MRTFA antibody has now been included in the Materials and Methods section under “Immunofluorescence.”

Reviewer #3

1. In a previous manuscript review, authors were asked to include primary fibroblasts to establish that, similarly to the embryonic cell line, they respond to ACLP. Instead, authors included results from adipose stromal vascular fraction cells, which did not improve this manuscript.

Please refer to our response to Reviewer #1.

March 20, 2026

RE: Life Science Alliance Manuscript #LSA-2025-03600-TRR

Dr. Matthew D Layne
Boston University
Biochemistry & Cell Biology
72 E. Concord St
Boston, MA 02118

Dear Dr. Layne,

Thank you for submitting your Research Article entitled "Aortic carboxypeptidase-like protein potentiates β 1 integrin signaling in mesenchymal progenitors". It is a pleasure to let you know that your manuscript is now accepted for publication in Life Science Alliance. Congratulations on this interesting work.

Your manuscript will now progress through copyediting and proofing. At the proofing stage, we request you to remove all sub-headings in the discussion section.

It is journal policy that authors provide original data upon request.

Your article will publish open access upon publication under a CC-BY license.

DISTRIBUTION OF MATERIALS:

Again, congratulations on a very nice paper. I hope you found the review process to be constructive and are pleased with how the manuscript was handled editorially. We look forward to future exciting submissions from your lab.

Sincerely,

Sarita Hebbar, PhD
Scientific Editor
Life Science Alliance
<http://www.lsajournal.org>